# Annexin A5 Inhibits Endothelial Inflammation Induced by Lipopolysaccharide-Activated Platelets and Microvesicles via Phosphatidylserine Binding

**DOI:** 10.3390/ph16060837

**Published:** 2023-06-03

**Authors:** Brent J. Tschirhart, Xiangru Lu, Janice Gomes, Arundhasa Chandrabalan, Gillian Bell, David A. Hess, Guangxin Xing, Hong Ling, Dylan Burger, Qingping Feng

**Affiliations:** 1Department of Physiology and Pharmacology, Schulich School of Medicine and Dentistry, Western University, London, ON N6A 5C1, Canada; 2Robarts Research Institute, Schulich School of Medicine and Dentistry, Western University, London, ON N6A 5C1, Canada; 3Department of Biochemistry, Schulich School of Medicine and Dentistry, Western University, London, ON N6A 5C1, Canada; 4Kidney Research Centre, Ottawa Hospital Research Institute, University of Ottawa, Ottawa, ON K1H 8L6, Canada

**Keywords:** sepsis, endotoxemia, inflammation, annexin A5, endothelial cells, platelets, extracellular vesicles

## Abstract

Sepsis is caused by a dysregulated immune response to infection and is a leading cause of mortality globally. To date, no specific therapeutics are available to treat the underlying septic response. We and others have shown that recombinant human annexin A5 (Anx5) treatment inhibits pro-inflammatory cytokine production and improves survival in rodent sepsis models. During sepsis, activated platelets release microvesicles (MVs) with externalization of phosphatidylserine to which Anx5 binds with high affinity. We hypothesized that recombinant human Anx5 blocks the pro-inflammatory response induced by activated platelets and MVs in vascular endothelial cells under septic conditions via phosphatidylserine binding. Our data show that treatment with wildtype Anx5 reduced the expression of inflammatory cytokines and adhesion molecules induced by lipopolysaccharide (LPS)-activated platelets or MVs in endothelial cells (*p* < 0.01), which was not observed with Anx5 mutant deficient in phosphatidylserine binding. In addition, wildtype Anx5 treatment, but not Anx5 mutant, improved trans-endothelial electrical resistance (*p* < 0.05) and reduced monocyte (*p* < 0.001) and platelet (*p* < 0.001) adhesion to vascular endothelial cells in septic conditions. In conclusion, recombinant human Anx5 inhibits endothelial inflammation induced by activated platelets and MVs in septic conditions via phosphatidylserine binding, which may contribute to its anti-inflammatory effects in the treatment of sepsis.

## 1. Introduction

Sepsis is a life-threatening organ dysfunction caused by a dysregulated host immune response to infection [1] and is a leading cause of mortality and morbidity among intensive care unit patients [2,3]. Adults 65 years or older, children less than one year old, people with weakened immune systems, and with chronic diseases such as diabetes, lung disease, cancer, and kidney disease are at higher risk of sepsis [4]. Currently, there is no specific effective treatment for the systemic inflammatory response other than targeting the source of infection and life support measures [1,5]. Vascular endothelial dysfunction and injury contribute to a sharp decline in blood pressure due to vascular leakage of blood components, known as extravasation, leading to ischemic and inflammatory organ damage, a characteristic of severe sepsis and septic shock [2,6].

Annexin A5 (Anx5) is a Ca^2+^ binding protein that is ubiquitously expressed in human tissues. Anx5 is secreted into the extracellular space and bloodstream at subtherapeutic levels [7,8]. A unique feature of Anx5 is its Ca^2+^-dependent high-binding affinity to phosphatidylserine to form a two-dimensional protein network on cell surfaces. The Anx5 protein network inhibits the cell’s interaction or communication with extracellular factors [8,9]. Quadruple mutation of Ca^2+^ binding sites on Anx5 (E72Q, D144N, E228Q, D303N), which is referred to as Anx5 mutant in this study, has been reported to abolish phosphatidylserine binding [10]. 

Anx5 binding on live cells may also act as a physical restraint to inhibit extracellular vesicle blebbing from host cells [11]. Anx5 binding to extracellular vesicles increases their uptake and reduces the circulating levels in *E. coli*-infected mice, ultimately delaying the onset of sepsis [12]. We and others have shown that Anx5 inhibits cytokines expression and improves organ function and animal survival in rodent models of sepsis [13,14]. Notably, Anx5 inhibits high mobility group box 1 (HMGB1)-mediated inflammation and the toll-like receptor 4 (TLR4) pathway activated by lipopolysaccharides (LPS) of Gram-negative bacteria [13,14,15].

Platelets fulfill their procoagulant role in hemostasis by externalizing phosphatidylserines, a process that is upregulated in sepsis [16,17]. Externalized phosphatidylserines modulate tissue factor activation, initiate the coagulation cascade, and promote inflammatory cytokine release [18,19]. Activated platelets are also able to “pinch” off portions of their membrane to form microvesicles (MVs) [20]. Through transporting their cargo, MVs promote the inflammatory activity of the target cells [21]. Notably, engineered nanoparticles are developed as drug-delivery vehicles for immunotherapy and infection control [22,23]. In septic conditions, the quantity of phosphatidylserine-positive MVs is elevated [24], contributing to inflammation, as the membranes of platelet-derived MVs are 50–100 times more pro-coagulant than those of activated platelets [25]. However, whether Anx5 inhibits inflammation induced by activated platelets and MVs via binding to the externalized phosphatidylserines in septic conditions is not well understood. We hypothesized that Axn5 blocks the pro-inflammatory response induced by LPS-activated platelets and MVs in vascular endothelial cells via phosphatidylserine binding.

## 2. Results

### 2.1. MV Morphology, Size, and Quantity in Endotoxemic Mice

To assess the morphology of the isolated MVs, transmission electron microscopic analysis was performed. A typical electron microscopic image of MV is shown in Figure 1A, in which the cellular components can be visualized by the dark shading within the structurally intact MV. The vesicle size was assessed using nanoparticle tracking analysis. The majority (~65%) of the extracellular vehicles isolated from control and endotoxemic mice ranged in size from 120–210 nm with negligible quantities over 500 nm (Figure 1B). The concentrations of circulating MVs from endotoxemic mice were 1.7-fold higher than those isolated from control mice (*p* < 0.01, Figure 1B). Notably, treatment with wildtype human recombinant Anx5 significantly lowered plasma MV levels induced by LPS (*p* < 0.01, Figure 1C).

### 2.2. In Vitro LPS Stimulation Leads to Higher Phosphatidylserine Exposure on Platelets and MVs

To investigate whether LPS stimulation increases phosphatidylserine exposure in platelets, mice were challenged with LPS (4 mg/kg, IP) for 4 h and the blood was collected for platelets isolation. Additionally, platelets were isolated from control mice and treated with LPS (1 µg/mL) for 4 h *in vitro*. The platelets and externalized phosphatidylserine were successfully labeled with CD41-AF647 and Anx5-FITC, respectively. The flow cytometric analysis showed that *in vivo* LPS stimulation did not significantly change platelet Anx5 positivity compared to control conditions. However, *in vitro* LPS stimulation increased Anx5^+^ platelets by 8.7- and 8.2-fold compared to control and *in vivo* LPS stimulation, respectively (*p* < 0.001, Figure 2A), indicating platelet activation following *in vitro* LPS stimulation. Similarly, more MVs were Anx5^+^ when they were stimulated by LPS *in vitro* (81%) compared to MVs isolated from control mice (68%) (*p* < 0.05, Figure 2B).

### 2.3. Anx5 Mutant Is Unable to Bind to Externalized Phosphatidylserine on LPS-Activated Platelets or MVs

To assess the binding of Anx5 quadruple mutant (E72Q, D144N, E228Q, D303N) to externalized phosphatidylserine, fluorescein-labeled wildtype and mutant Anx5 were incubated with the LPS stimulated platelets and MVs. Flow (platelets) and nanoflow (MVs) cytometric analyses show that 91.9% of platelets and 80.6% of MVs were labeled with wildtype Anx5, whereas mutant Anx5 labeled only 4.9% and 0.1% of the platelets and MVs, respectively (*p* < 0.001, Figure 3A,B). 

### 2.4. Anx5 Attenuates Pro-Inflammatory and Pro-Adhesive Responses of LPS-Stimulated Platelets and MVs via Phosphatidylserine Binding

To examine the anti-inflammatory and anti-adhesive effects of Anx5, cultured microvascular endothelial cells were incubated with LPS-stimulated platelets or MVs in the presence or absence of wildtype or mutant Anx5 in cell media for 4 h. LPS-stimulated platelets induced significantly higher levels of inflammatory cytokine (TNFα and IL-6) and adhesion molecule (E-selectin and ICAM-1) expression in endothelial cells compared to the controls (*p* < 0.001, Figure 4A). Wildtype Anx5 (1 µg/mL) alone had no effect on cytokine levels but significantly lowered inflammatory cytokine and adhesion molecule expression induced by LPS-stimulated platelets (*p* < 0.001, Figure 4A). The mutant Anx5 had no inhibitory effect on the expression of both inflammatory cytokines and adhesion molecules induced by LPS-stimulated platelets (Figure 4A).

Similarly, LPS-stimulated MVs induced significantly higher expression of TNFα and IL-6, E-selectin, and ICAM-1 in endothelial cells compared to the controls (*p* < 0.001, Figure 4B). Wildtype Anx5 (1 µg/mL) alone had no effect on cytokine levels; however, the inflammatory cytokine and adhesion molecule expression induced by LPS-stimulated MVs was significantly downregulated by wildtype Anx5 (*p* < 0.001), but not by the mutant Anx5 (Figure 4B).

### 2.5. Trans-Endothelial Electrical Resistance (TEER) in Septic Conditions

Microvascular endothelial cells were cultured to reach confluence in trans-well inserts with an average baseline TEER reading of 200.5 ± 2.8 Ω, indicating endothelial monolayer integrity (100%). In response to LPS, LPS-stimulated platelets, or LPS-stimulated MVs, TEER gradually reduced below 25% of their baseline values over the course of 24 h with the largest reduction occurring at the 30 min timepoint (Figure 5A–C). Treatment with wildtype Anx5 led to significantly higher TEER values at various timepoints over the course of 24 h (*p* < 0.01). Notably, treatment with the mutant Anx5 did not increase TEER values in endothelial monolayers incubated with LPS-stimulated platelets or MVs (Figure 5B,C).

### 2.6. Anx5 Inhibits LPS-Induced Monocyte Adhesion to Endothelial Cells

To complement our findings that Anx5 reduces adhesion molecule expression, LPS-induced monocyte adhesion to cultured microvascular endothelial cells was assessed with and without wildtype or mutant Anx5 (Figure 6A). Quantitative analysis showed that treatment with wildtype or mutant Anx5 (1 µg/mL) alone did not significantly alter monocyte adhesion compared to controls. Whereas LPS (1 µg/mL) challenge significantly increased monocyte adhesion to the endothelial monolayer (*p* < 0.001), which was significantly inhibited by wildtype Anx5 (*p* < 0.001) but not by the quadruple mutant of Anx5 (Figure 6B).

### 2.7. Anx5 Inhibits LPS-Induced Platelet Adhesion to Endothelial Cells

To examine the effects of Anx5 on platelet adhesion, LPS-induced platelet adhesion to cultured microvascular endothelial cells was assessed with and without wildtype or quadruple mutant of Anx5. Representative images are shown in Figure 7A. As observed with monocytes, treatment with wildtype or mutant Anx5 (1 µg/mL) alone did not significantly alter platelet adhesion to endothelial cells compared to controls. Whereas LPS (1 µg/mL) challenge significantly increased platelet adhesion to the endothelial monolayer (*p* < 0.01), which was significantly inhibited by wildtype Anx5 (*p* < 0.001) but not by the mutant Anx5 (Figure 7B). 

## 3. Discussion

The main finding of the present study is that wildtype but not the phosphatidylserine binding-deficient mutant of Anx5 inhibited inflammatory cytokine and adhesion molecule expression induced by LPS-stimulated platelets or MVs in microvascular endothelial cells. We also demonstrated that wildtype but not the mutant Anx5 attenuated the disruption of endothelial monolayer integrity induced by LPS-stimulated platelets or MVs. Furthermore, we showed that the wildtype but not mutant Anx5 diminished LPS-induced platelet and monocyte adhesion to microvascular endothelial cells. Overall, our study suggests that the phosphatidylserine binding of Anx5 is critical to its anti-inflammatory and anti-adhesive effects in septic conditions.

Under normal physiological conditions, phosphatidylserine is contained within the inner leaflet of the cell membrane [26,27]. An enzyme known as TMEM16F, a phospholipid scramblase, is involved in the process of externalizing phosphatidylserine in a Ca^2+^-dependent manner [28,29]. Additionally, LPS promotes phosphatidylserine exposure through the formation of gasdermin pores in platelets and macrophages [30]. As phosphatidylserine is externalized, it creates a pro-inflammatory and pro-coagulant surface on the cell [16,31]. Previous studies have shown that phosphatidylserine exposure is required for activating the “sheddase” ADAM17 which cleaves TNFα and IL-6R along with other inflammatory cytokine and chemokine expression to promote inflammation [19,32]. In the present study, *in vitro* but not *in vivo* LPS stimulation significantly increased Anx5^+^ platelets, indicating phosphatidylserine externalization of platelets *in vitro* but not *in vivo* conditions. This finding is consistent with recent studies showing that phosphatidylserine externalization mediates the clearance of activated platelets by the vascular endothelium during sepsis *in vivo* [17]. Additionally, platelets with phosphatidylserine exposure can be phagocytosed by macrophages [33]. It is possible that during *in vivo* LPS stimulation, platelets with externalized phosphatidylserine are activated and aggregated, or bind to the microvascular endothelium [34], impeding our ability to obtain these activated platelets through blood collection and platelet isolation. 

Platelets express toll-like receptors including TLR4. In response to LPS stimulation, platelets are activated [35,36]. Activated platelets release cytokines including IL-1 and damage-associated molecular patterns (DAMPs), such as high mobility group box 1 (HMGB1), and expose glycoproteins including P-selectin and CD40 ligands to their surface, promoting interaction with immune cells and platelet adhesion to the vascular endothelium [37]. Furthermore, platelets foster endothelial adhesion and extravasation of leukocytes, enhancing cytokine expression and the pro-inflammatory response in sepsis [38]. However, the underlying molecular mechanisms by which platelets induce inflammation are still not fully understood. In the present study, we tested the hypothesis that phosphatidylserine exposure of the activated platelets mediates endothelial inflammation in septic conditions. We found that Anx5 inhibited endothelial inflammatory cytokine (TNFα and IL-6) and adhesion molecule (E-selectin and ICAM-1) expression induced by activated platelets. Additionally, Anx5 impeded platelet and monocyte adhesion to vascular endothelial cells in septic conditions. Notably, these inhibitory effects of Anx5 were abrogated by a quadruple mutation of Anx5 in which four Ca^2+^ binding sites are mutated (E72Q, D144N, E228Q, and D303N) to prevent it from binding to phosphatidylserine as Anx5 binding to phosphatidylserine is Ca^2+^-dependent [10]. These results suggest that phosphatidylserine exposure mediates endothelial inflammation induced by activated platelets in septic conditions.

During sepsis, circulating MVs are elevated, which are derived from several cellular sources, including activated platelets, white blood cells, and endothelial cells [20,24]. The majority of the MVs have phosphatidylserine exposure on the vesicle surface [39,40]. In the present study, following LPS stimulation, 81% of the MVs were phosphatidylserine-positive. Previous studies have shown that MVs promote inflammation through the production of reactive oxygen species (ROS), the release of cytokines, activation of nuclear factor-kappa B (NF-κB), and recruitment of immune cells to the site of injury [20,41,42]. However, the role of phosphatidylserine exposure in inflammation induced by MVs is not fully elucidated. In the present study, we showed that Anx5 inhibited endothelial inflammatory cytokine (TNFα and IL-6) and adhesion molecule (E-selectin and ICAM-1) expression induced by LPS-stimulated MVs. As sepsis develops, the structural integrity of the endothelial monolayer is compromised, resulting in vascular leakage and trans-endothelial migration of monocytes and neutrophils promoting inflammation [43]. To assess endothelial monolayer integrity, we monitored TEER and found that Anx5 treatment prevented the reduction in TEER-induced LPS-stimulated platelets and MVs. Strikingly, the inhibitory effects of Anx5 were abrogated by a quadruple mutation of Anx5, indicating an important role of phosphatidylserine exposure in the inflammatory response induced by platelets and MVs during septic conditions.

## 4. Methods

### 4.1. Experimental Animals and Ethics

Adult male C57BL/6J mice weighing 20–25 g were purchased from Charles River Laboratories (Laval, Canada), and housed at 24 ± 4 °C in a well-ventilated animal care facility under a 12:12 h light and dark cycle. Mice had free access to standard mouse chow and water *ad libitum*. The animals were acclimatized in the animal care facility for one week prior to the start of the experiment. All animals received humane care. Animal procedures were performed in accordance with the Guide to Care and Use of Animals of the Canadian Council of Animal Care (CCAC), including the 3Rs concept, and were approved by the Animal Care Committee at Western University, Canada (Animal Use Protocol number AUP-2020-128; Approval Date, 17 March 2021). All methods were carried out to minimize the number of animals used and their suffering. 

### 4.2. In Vivo Endotoxemia 

LPS (Salmonella Tryphosa, Cat. L7136, Sigma Aldrich, St. Louis, MO, USA) was used to induce endotoxemia on C57BL/6 mice via an intraperitoneal (IP) injection at 4 mg/kg body weight. Blood was collected for platelet and MV isolation, 4 h after LPS injection.

### 4.3. Mouse Skeletal Muscle Endothelial Cell Culture 

Primary microvascular endothelial cells were isolated from C57BL/6 mice (about 2 months of age). Mice were euthanized and hindlimb skeletal muscles were digested in a 15 mL digestion buffer containing collagenase II (1 mg/mL), dispase II (1.6 mg/mL), and fetal bovine albumin (1.6 mg/mL) in DMEM medium at 37 °C for 40 min. Digested tissues were dissociated by agitations using an electronic pipette followed by centrifugation at 300 g for 5 min. The pellet was resuspended in PBS and strained using a 70 µm filter. Cells were collected by centrifugation and resuspended in DMEM medium containing 20% fetal bovine serum, 1% penicillin/streptomycin, 1% L-glutamine, microvascular growth supplement (Thermo Scientific, Waltham, MA, USA), and 5 U/mL heparin. Cultured endothelial cells were labeled with fluorescein-isolectin B4 (Vector Laboratory, Burlington, ON, Canada) to visualize cell purity, which was consistently ~98%.

### 4.4. Platelet and Microvesicle Isolation 

Platelets and MVs were isolated following a modified protocol from Burger et al. [41]. Briefly, adult C57BL/6 mice were sedated using intraperitoneal injections of ketamine (150 mg/kg), xylazine (10 mg/kg), and acepromazine (5 mg/kg). Blood was obtained through cardiac puncture using a 23-gauge needle connected to a heparinized syringe and transferred to a heparinized tube. Blood was spun at 180 g at 4 °C for 8 min to obtain platelet-rich plasma. Plasma was then spun at 800× *g* (4 °C) for 10 min to obtain a platelet pellet and the supernatant was spun at 19,500× *g* at 4 °C for 20 min to obtain an MV pellet. The isolated platelets were quantified using a hemocytometer (Bright-Line) under a light microscope. Ten million platelets were used for each experiment condition.

### 4.5. Incubation of In Vitro LPS-Stimulated Platelets or MVs with Cultured Endothelial Cells

Isolated platelets and MVs were suspended in PBS. LPS was added at a concentration of 1 µg/mL for 4 h. After LPS treatment, the platelets and MVs were washed two times with PBS. The pellets were resuspended with DMED medium and added to the cultured endothelial cells in 24-well plates and incubated for 4 h at 37 °C containing 5% CO_2_. Endothelial cells treated with LPS (1 µg/mL) alone for 4 h were used as a positive control. After 4 h of treatment, the culture medium was removed, and cells were washed with PBS three times. RNAs isolated from the treated endothelial cells were used for the assessment of inflammatory cytokine and adhesion molecule mRNA levels via RT-qPCR analysis. 

### 4.6. Recombinant Human Wildtype and Quadruple Mutant of Anx5

Wildtype human Anx5 and its quadruple mutant (E72Q, D144N, E228Q, and D303N) proteins were recombinantly produced. Briefly, the DNA sequences of wildtype and quadruple mutant of Anx5 were codon optimized, synthesized, and cloned into pEX-N-His expression vector (PS100030, Origene, Rockville, MD, USA). The plasmids were transformed into BL21(DE3)-competent cells to produce recombinant proteins, which were purified using His-tag affinity chromatography (Nickel column, HisTrap HP^TM^, GE Healthcare, Chicago, IL, USA). The His-tags were cleaved from the purified protein samples. Endotoxin was removed using a ToxinEraser™ endotoxin removal kit (GenScript, Piscataway, NJ, USA). The concentrations of recombinant proteins were determined using Bradford assay.

### 4.7. Nanoparticle Tracking Analysis of Extracellular Vesicles

MVs isolated from C57BL/6 mice were treated with or without LPS (4 mg/kg, IP) for 4 h, washed and suspended in 100 µL PBS, and frozen at −80 °C until analysis. To assess extracellular vesicle size and quantity, samples were further diluted to a final volume of 20 mL. The diluted EV sample (1 mL) was loaded into a nanoparticle tracking analysis machine (ZetaView PMX110, Particle Metrix, Meerbusch, Germany) after calibration using 105 nm and 500 nm polystyrene beads. ZetaView software (version 8.02.28) was used for the analysis of light scattering at 11 camera positions with 2 s video lengths, a camera frame rate of 15 frames per second, with a system temperature of ~21 °C to obtain the size profiling and quantification of isolated extracellular vesicles.

### 4.8. Transmission Electron Microscopic Analysis 

MVs isolated from C57BL/6 mice were suspended in 100 µL PBS and frozen at −80 °C. MV pellets were fixed with 2.5% glutaraldehyde in PBS for 4 hours at room temperature. Next, the pellet was washed in 0.1 M Na cacodylate buffer, post-fixed in 2% OsO4 and dehydrated in a series of graded ethanol dilutions. Samples were embedded in Spurr Resin, and 60 nm sections were prepared on copper grids. Samples were loaded onto Formvar/carbon-coated copper grinds. Images were captured using a JEOL JEM-1400 plus transmission electron microscope (Akishima, Tokyo, Japan) [44].

### 4.9. Nanoflow Cytometric Analysis

MVs were analyzed using the A50-Micro plus nanoscale flow cytometer (Apogee Flow Systems Ltd., Northwood, UK) as previously described [45]. Light scatter triggering thresholds were determined with the smallest particles distinguishable from noise (110 nm polystyrene beads). Thresholds were set at 20 a.u. small angle light scatter (SALS) and 25 a.u. large angle light scatter (LALS). Cellular fragments within the size range of ~180–880 nm were classified as MVs. Control and LPS-stimulated samples were labeled with FITC-conjugated Anx5 (BioLegend, San Diego, CA, USA) and fluorescein-5-maleimide-conjugated Anx5 mutant (E72Q, D144N, E228Q, and D303N; deficient in phosphatidylserine binding) [10], separately. The mutant Anx5 was labeled using a sulfhydryl-reactive fluorescein-5-maleimide (Thermo Scientific, Waltham, MA, USA) in-house. The events of Anx5-labeled MVs and total MVs in the 180–880 nm size range were counted. The percent of FITC-Anx5 or fluorescein-Anx5 mutant positive MVs to total MVs was determined.

### 4.10. Platelet Flow Cytometry

Freshly isolated mouse platelets were stained for CD41-AF674 (1 mL/sample, BioLegend, San Diego, CA, USA), wildtype Anx5-FITC (2 µL/sample), and mutant Anx5-fluorescein (2 mL/sample) at a concentration of 1 million platelets in 200 mL annexin-binding buffer for 30 min at room temperature. CD41 and Anx5 expressions were determined using an LSRII flow cytometer (BD Biosciences, Franklin Lakes, NJ, USA) at the London Regional Flow Cytometry Facility, Western University. Platelets were first identified as CD41^+^ events and the percentage of CD41^+^ events that were Anx5^+^ was quantified using FlowJo software v.8.2.

### 4.11. RT-qPCR Analysis

Total RNAs isolated from cultured microvascular endothelial cells were reverse transcribed to cDNA using M-MLV reverse transcriptase (Thermo Scientific, Waltham, MA, USA). The cDNA was amplified in the presence of EvaGreen qPCR Mastermix (Applied Biological Materials Inc., Richmond, BC, Canada) containing gene-specific primers using an Eppendorf Realplex^2^ MasterCycler (Hamburg, Germany). All samples were amplified for 35 cycles and the cycle threshold (Ct) of each sample was determined using Realplex software. The mRNA levels in relation to 28S ribosomal RNA as an internal control were determined using a comparative CT method. Primer sequences (5′-3′) of genes (GenBank sequence ID) used for the qPCR analysis were: 
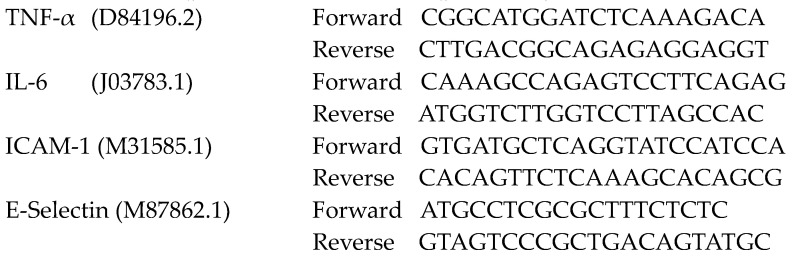


### 4.12. Trans-Endothelial Electrical Resistance (TEER)

Endothelial cells were cultured onto transwell inserts (1 µm pore, Greiner Bio-One, Kremsmünster, Austria) and grown to 100% confluence. TEER was determined using an epithelial volt/ohm meter 3 (EVOM3, World Precision Instruments, Sarasota, FL, USA) with STX2-plus electrodes. Experiments were started after TEER values plateaued. TEER was assessed every 30 min for 4 h and then at 8 and 24 h following each treatment. In each experiment, TEER values from wells without any treatment were used as controls. Percentage change to the baseline was calculated for each timepoint.

### 4.13. Monocyte Isolation and Adhesion Assay

C57BL/6 mice were euthanized and monocytes were obtained from femoral bone marrow. Briefly, both hind limbs were surgically removed, and the femur was cut at both epiphyses, exposing the bone marrow. A 1 mL syringe with a 26-gauge needle was inserted into the bone marrow cavity to flush out bone marrow using PBS. Suspended bone marrow was filtered using a nylon filter to remove any non-cellular debris before centrifugation at 250 g for 10 min to obtain a pellet consisting of bone marrow cells. Red blood cells were removed using a red blood cell lysis buffer. On average, ~12 million monocytes were collected per mouse.

Mouse microvascular endothelial cells were passaged onto a 96-well plate and grown to 100% confluence. Cells were treated with LPS (1 µg/mL) in the presence or absence of wildtype or quadruple mutant of Anx5 (1 µg/mL) for 3 h followed by incubation with isolated monocytes from wildtype mice for 1 h at 37 °C under static conditions. The media was then removed. Cells were gently washed 3 times with PBS to remove unadhered monocytes. The adhered monocytes to cultured endothelial cells were visualized and quantified from images of random fields under a light microscope using Zen 3.4 software. Values were expressed as fold change in adhered monocytes vs. the control.

### 4.14. Platelet Adhesion Assay 

Mouse microvascular endothelial cells were cultured on cover glass to confluence and treated with LPS (1 µg/mL) in the presence or absence of wildtype or quadruple mutant of Anx5 (1 µg/mL) for 3 h followed by incubation with calcein-AM-labeled platelets from wildtype mice for 1 h. The media was removed, and cells were gently washed 3 times with PBS. Cells were then fixed with 4% paraformaldehyde (PFA), stained with Hoechst, and mounted on slides before visualization via fluorescence microscope. The adhered platelets were counted per random field using AxioVision V4.8.2 software. Results were normalized to the average platelet count of each replicate’s control and expressed as fold change vs. the control.

### 4.15. Statistical Analysis

Data are presented as mean ± SEM. An unpaired Student’s t-test was utilized to compare the difference between two groups. One-way analysis of variance (ANOVA) followed by Fisher’s LSD or Tukey’s test was used for multiple group comparisons (Prism 9, GraphPad, San Diego, CA, USA).

## 5. Conclusions

In this present study, we showed that treatment with wildtype Anx5 lowered the expression of inflammatory cytokines and adhesion molecules induced by LPS-activated platelets or MVs in vascular endothelial cells through phosphatidylserine binding. In addition, Anx5 treatment reduced monocyte and platelet adhesion to endothelial cells and improved vascular endothelial integrity in septic conditions. Our study suggests that Anx5 has great potential in the treatment of sepsis. A pilot randomized trial on the feasibility and safety of recombinant human Anx5 in severe COVID-19 with sepsis was completed (NCT04748757). A phase 2 randomized trial to assess the efficacy and safety of recombinant human Anx5 in bacterial sepsis is ongoing (NCT04898322). Results from these trials may help to better understand the therapeutic potential of Anx5 in patients with sepsis.

## Figures and Tables

**Figure 1 pharmaceuticals-16-00837-f001:**
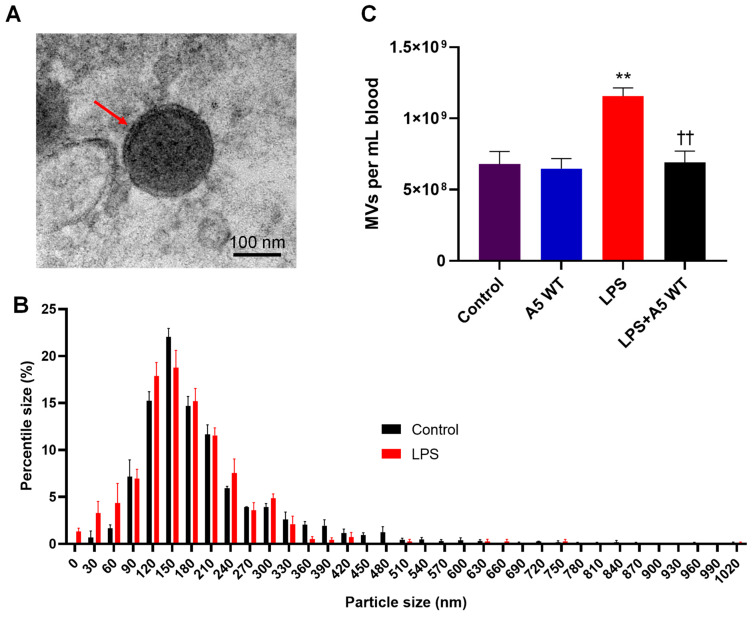
**Morphology, size, and concentration of microvesicles (MVs).** (**A**) Transmission electron microscope image. The red arrow points to a structurally intact MV. (**B**) Size distribution histogram of extracellular vesicles analyzed by nanoparticle tracking analysis. (**C**) Effects of LPS (4 mg/kg, IP) without and with wildtype Anx5 (10 µg/kg, IV) treatment on circulating MV concentrations in mice in vivo. Data are mean ± SEM from *n* = 3–5 mice per group. One-way ANOVA with Tukey’s test was used for statistical analysis. ** *p* < 0.01 vs. control, †† *p* < 0.01 vs. LPS.

**Figure 2 pharmaceuticals-16-00837-f002:**
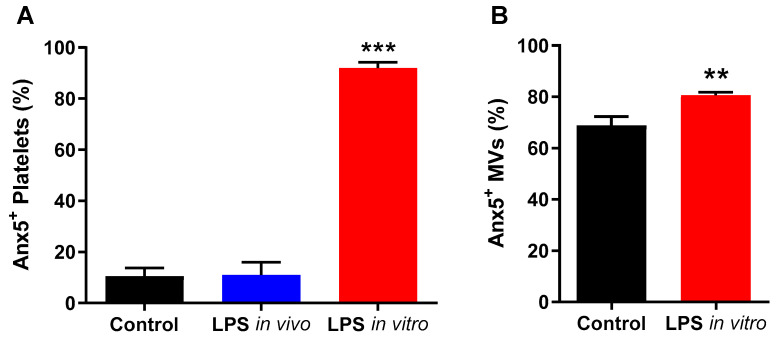
***In vitro* LPS stimulation increases phosphatidylserine exposure in platelets and MVs.** Platelets and microvesicles (MVs) were isolated from mouse blood collection. (**A**) Platelets were isolated from C57BL/6 mice (control) and stimulated with LPS stimulated (1 µg/mL) *in vitro* or from mice treated with LPS (4 mg/kg, IP) *in vivo*. (**B**) Quantification of Anx5^+^ MVs isolated from plasma of C57BL/6 mice without (control) and with LPS (1 µg/mL) stimulation *in vitro*. Data are mean ± SEM from *n* = 4–5 mice. One-way ANOVA followed by Tukey’s test in (**A**) and unpaired Student’s *t*-test in (**B**) were used for statistical analysis. *** *p* < 0.001 vs. other 2 groups (**A**). ** *p* < 0.01 vs. control (**B**).

**Figure 3 pharmaceuticals-16-00837-f003:**
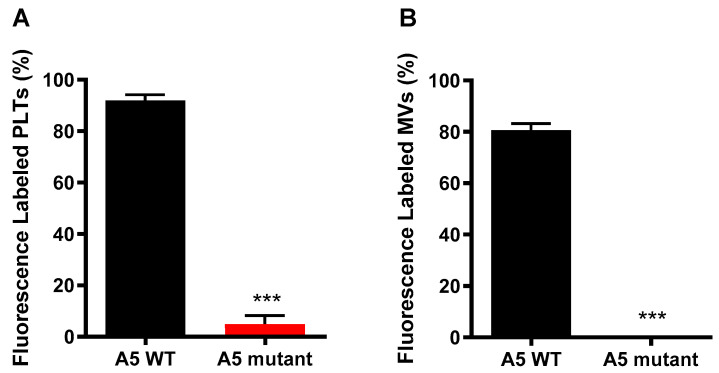
**Quadruple mutant of Anx5 (A5 mutant) is deficient in binding to phosphatidylserine on LPS-activated platelets or MVs.** Platelets (PLTs) and microvesicles (MVs) were isolated from mouse blood collection. PLTs (**A**) and MVs (**B**) were stimulated with 1 µg/mL LPS *in vitro* and labeled with FITC-conjugated wildtype Anx5 (WT A5) or fluorescein-conjugated quadruple mutant of Anx5 (A5 mutant). Data are mean ± SEM from *n* = 5 mice, *** *p* < 0.001 vs. A5 WT by unpaired Student’s *t*-test.

**Figure 4 pharmaceuticals-16-00837-f004:**
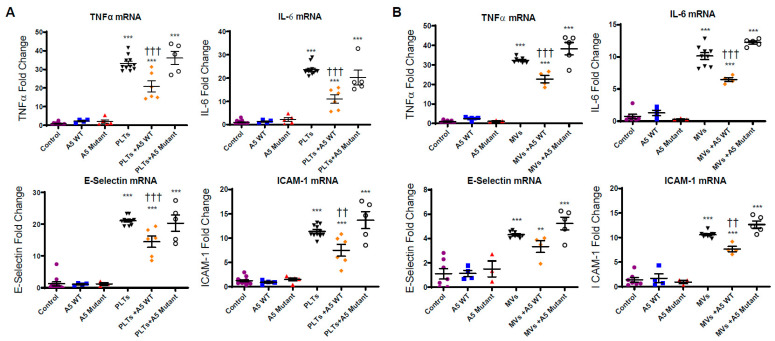
**Effects of wildtype (A5 WT) or quadruple mutant of Anx5 (A5 mutant) on mRNA levels of cytokines and adhesion molecules in cultured microvascular endothelial cells induced by LPS-stimulated platelets (PLTs) or MVs.** Cultured microvascular endothelial cells were incubated with LPS-stimulated PLTs (**A**) or MVs (**B**) in the absence or presence of A5 WT or mutant (1 µg/mL) for 4 h. The mRNA levels in cultured microvascular endothelial cells were analyzed by RT-qPCR. Data are mean ± SEM from *n* = 4–7 independent cell cultures per group. One-way ANOVA followed by Tukey’s multiple comparison test was used for statistical analysis. ** *p* < 0.01. *** *p* < 0.001 vs. control. †† *p* < 0.01, ††† *p* < 0.001 vs. LPS-stimulated PLTs or MVs alone.

**Figure 5 pharmaceuticals-16-00837-f005:**
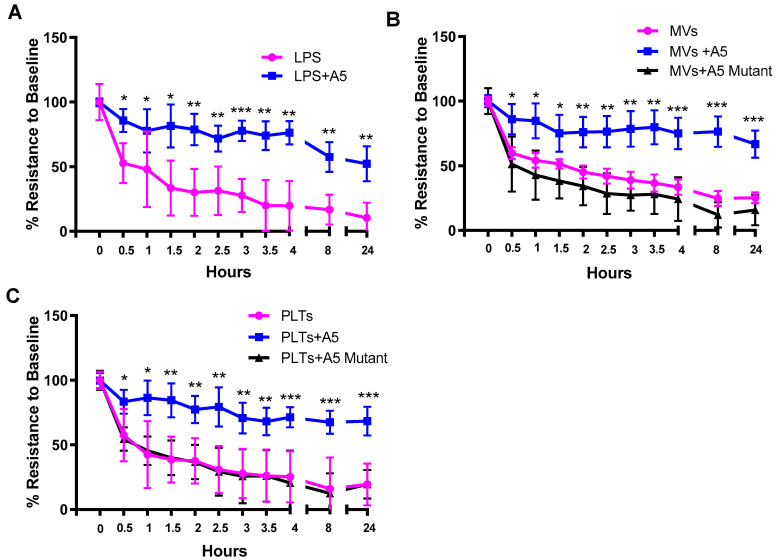
**Effects of wildtype (A5 WT) or quadruple mutant of Anx5 (A5 mutant) on trans-endothelial electrical resistance (TEER) induced by LPS, LPS-stimulated PLTS, and LPS-stimulated MVs.** The confluent endothelial cells were challenged with LPS (4 µg/mL) (**A**), LPS-stimulated PLTs (**B**), or LPS-stimulated MVs (**C**) in the absence or presence of A5 WT or mutant (1 µg/mL). TEER was determined using an EVOM3 meter. Data are mean ± SEM from four independent cultures. Two-way ANOVA followed by Fisher’s LSD test was used for statistical analysis. * *p* < 0.05, ** *p* < 0.01, *** *p* < 0.001 vs. LPS, or the other two groups.

**Figure 6 pharmaceuticals-16-00837-f006:**
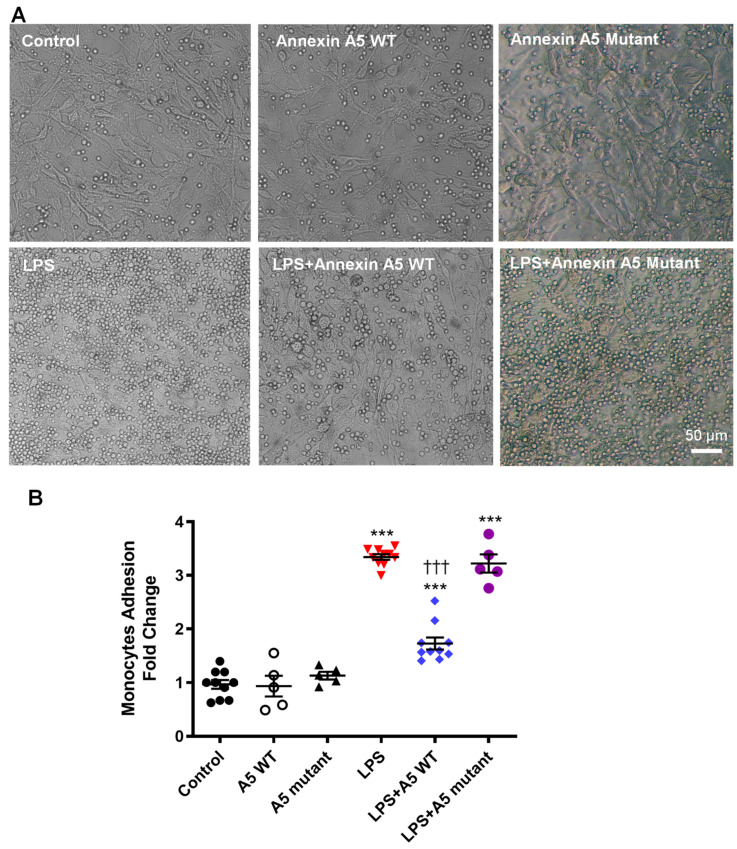
**Monocyte Adhesion to Endothelial Cells.** (**A**) Representative images of monocyte adhesion to cultured microvascular endothelial cells. The rounded cells are monocytes. (**B**) Quantification of monocyte adhesion to cultured microvascular endothelial cells in response to LPS (1 µg/mL), LPS (1 µg/mL) + A5 WT (1 µg/mL), and LPS (1 µg/mL) + A5 mutant (1 µg/mL). Data are mean ± SEM, and were analyzed using one-way ANOVA followed by Tukey’s multiple comparison test. *n* = 5–10 of independent cultures per group. *** *p* < 0.001 vs. control. ††† *p* < 0.001 vs. LPS alone. A5 WT and mutant indicate wildtype and quadruple mutant of annexin A5, respectively.

**Figure 7 pharmaceuticals-16-00837-f007:**
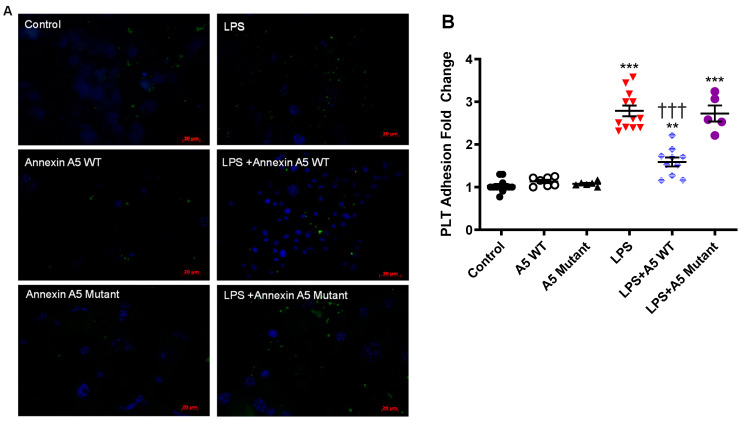
**Platelet adhesion to endothelial cells.** (**A**) Representative images of platelet adhesion to cultured microvascular endothelial cells. Green and blue colors indicate platelets and nuclear Hoechst staining, respectively. (**B**) Quantification of platelet adhesion to cultured microvascular endothelial cells in response to LPS (1 µg/mL), LPS (1 µg/mL) + A5 WT (1 µg/mL), and LPS (1 µg/mL) + A5 mutant (1 µg/mL). Results were normalized to the average platelet count to its replicate control. Data are mean ± SEM and were analyzed using one-way ANOVA followed by Tukey’s multiple comparison test. *n* = 5–10 of independent cultures per group. ** *p* < 0.01, *** *p* < 0.001 vs. control. ††† *p* < 0.001 vs. LPS alone.

## Data Availability

The datasets generated and/or analyzed during the current study are available from the corresponding author upon reasonable request.

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
