# Peer review of "Annexin A5 Inhibits Endothelial Inflammation Induced by Lipopolysaccharide-Activated Platelets and Microvesicles via Phosphatidylserine Binding"

_pharmaceuticals, 2023, doi:10.3390/ph16060837_

Round 1

Reviewer 1 Report

Tschirhart et al has shown that treatment with wild-type Anx5 reduced the expression of inflammatory cytokines and adhesion molecules induced by lipopolysaccharide (LPS)-activated platelets or MVs in endothelial cells, which was not observed with Anx5 mutant deficient in phosphatidylserine binding.

I fail to understand how this is relevant to in vivo condition if this cannot be detected in vivo. Is it possible to perform in vivo staining or some other method to prove the relevance. Otherwise the work is presented beautifully and clearly.

Author Response

Thank you for your favorable comments.

For microvesicles (MVs), we did observe a significant increase in MV concentrations in blood after in vivo LPS treatment in mice (Fig. 1C). For platelets, the percentage of Anx5 positive platelets was significantly increased only after LPS stimulation in vitro but not in vivo (Fig. 2A). It is possible that during in vivo LPS stimulation, platelets with externalized phosphatidylserine are activated and aggregated, or bind to the microvascular endothelium, or phagocytosed by macrophages, ultimately impeding our ability to obtain these activated platelets through blood collection and platelet isolation. This is explained in the Discussion section (lines 289-298).

Reviewer 2 Report

Dear authors

This manuscript has evaluated the effects of Annexin 5 anti-inflammatory effects caused by Lipopolysaccharide-Activated Platelets.

There are following comments for this manuscript:

1-      There is no similar previous study and therefore, this is a novel survey. The authors have published a related conference abstract (https://www.ahajournals.org/doi/abs/10.1161/circ.144.suppl_1.14208).

2-      The title and abstract of the manuscript are appropriate and relative to the text. However, more details of methods is needed in the abstract. If any statistical analysis has been also performed, please add p values of main findings in the abstract.

3-      The introduction section can be improved in the aspect of risk factors. Please also briefly describe Anx5 wildtype and mutants, mechanisms and effects/clinical outcomes in the introduction section.

4-      Methods is suitable. However, if you have received ethical code, please add in this section.

5-      Please add p-value scores in addition to percentages in parentheses.  

6-      Please write in vitro and in vivo in italic form.

7-      The writing of the manuscript is appropriate and needs minor revision.

8-      Results are appropriate; it is interesting that this role of Anx5 has been introduced for the first time without previous studies.

9-       Please follow journal style throughout the manuscript.  

1   Please use more related studies from 2022 in the discussion section.

1  As this topic is mainly on inflammation and also nanoparticles, the authors can use these titles of studies, published in MDPI journals, as references in the introduction “Metallic Nanoparticles: Their Potential Role in Breast Cancer Immunotherapy via Trained Immunity Provocation” and “Nickel Nanoparticles: Applications and Antimicrobial Role against Methicillin-Resistant Staphylococcus aureus Infections”.  

 With best regards

Dear authors

The English level can be improved

Author Response

Thank you for your careful review and favorable comments.

1-      There is no similar previous study and therefore, this is a novel survey. The authors have published a related conference abstract (https://www.ahajournals.org/doi/abs/10.1161/circ.144.suppl_1.14208).

Response: We are pleased to hear our study is novel. The published abstract is now acknowledged and cited.

2-      The title and abstract of the manuscript are appropriate and relative to the text. However, more details of methods is needed in the abstract. If any statistical analysis has been also performed, please add values of main findings in the abstract.

Response: We have included trans-endothelial electrical resistance measurements and p values in the revised abstract.

3-      The introduction section can be improved in the aspect of risk factors. Please also briefly describe Anx5 wildtype and mutants, mechanisms and effects/clinical outcomes in the introduction section.

Response: Risk factors for sepsis are now included in the Introduction section (lines 38-40). Anx5 mutant is defined and additional molecular mechanisms and effects of wildtype Anx5 are included in the Introduction section (lines 52, 58-59). 

4-      Methods is suitable. However, if you have received ethical code, please add in this section.

Response: The approved animal ethics code is included on line 346.

5-      Please add p-value scores in addition to percentages in parentheses.  

Response: Done.

6-      Please write in vitro and in vivo in italic form.

Response: Done.

7-      The writing of the manuscript is appropriate and needs minor revision.

Response: Revised accordingly.

8-      Results are appropriate; it is interesting that this role of Anx5 has been introduced for the first time without previous studies.

Response: Thank you for your favorable comments.

9-       Please follow journal style throughout the manuscript.  

Response: Done.

  • Please use more related studies from 2022 in the discussion section.

Response: Five additional related papers (Ref. 27, 29, 31, 39, and 40) from 2018 to 2023 are included in the Discussion section.

  • As this topic is mainly on inflammation and also nanoparticles, the authors can use these titles of studies, published in MDPI journals, as references in the introduction “Metallic Nanoparticles: Their Potential Role in Breast Cancer Immunotherapy via Trained Immunity Provocation” and “Nickel Nanoparticles: Applications and Antimicrobial Role against Methicillin-Resistant Staphylococcus aureus Infections”.  

Response: The suggested papers are cited in the Introduction as references #22 and #23.

With best regards

Comments on the Quality of English Language

The English level can be improved.

Response: Revised accordingly.

Round 2

Reviewer 1 Report

Thank you for the response. I agree that you have shown an increase in MVs in mice. However, I suggest you try staining perhaps to see this clearly. Your explanation is a mere hypothesis or an assumption and not supported by any scientific evidence.